# Design and Synthesis of (*Z*)-5-(Substituted benzylidene)-3-cyclohexyl-2-thioxothiazolidin-4-one Analogues as Anti-Tyrosinase and Antioxidant Compounds: In Vitro and In Silico Insights

**DOI:** 10.3390/antiox11101918

**Published:** 2022-09-27

**Authors:** Jeongin Ko, Jieun Lee, Hee Jin Jung, Sultan Ullah, Yeongmu Jeong, Sojeong Hong, Min Kyung Kang, Yu Jung Park, YeJi Hwang, Dongwan Kang, Yujin Park, Pusoon Chun, Jin-Wook Yoo, Hae Young Chung, Hyung Ryong Moon

**Affiliations:** 1Department of Manufacturing Pharmacy, College of Pharmacy, Pusan National University, Busan 46241, Korea; 2Department of Pharmacy, College of Pharmacy, Pusan National University, Busan 46241, Korea; 3Department of Molecular Medicine, The Scripps Research Institute, Jupiter, FL 33458, USA; 4New Drug Development Center, Department of Medicinal Chemistry, Daegu-Gyeongbuk Medical Innovation Foundation, Daegu 41061, Korea; 5College of Pharmacy and Inje Institute of Pharmaceutical Sciences and Research, Inje University, Gimhae 50834, Korea

**Keywords:** tyrosinase, PUSC scaffold, antioxidant, anti-melanogenesis, docking, rhodanine

## Abstract

Many compounds containing the β-phenyl-α,β-unsaturated carbonyl (PUSC) scaffold, including cinnamamide derivatives, have been shown to inhibit tyrosinase potently in vitro and in vivo. Structural changes to cinnamamide derivatives were produced by adding a dithionate functional group to provide eight (*Z*)-5-(substituted benzylidene)-3-cyclohexyl-2-thioxothiazolidin-4-one analogs with high log *p* values for skin. These analogs were synthesized using a two-step reaction, and their stereochemistry was confirmed using the ^3^*J*_C4-Hβ_ values of C4 measured in proton-coupled ^13^C mode. Analogs **2** (IC_50_ = 5.21 ± 0.86 µM) and **3** (IC_50_ = 1.03 ± 0.14 µM) more potently inhibited mushroom tyrosinase than kojic acid (IC_50_ = 25.26 ± 1.10 µM). Docking results showed **2** binds strongly to the active site of tyrosinase, while **3** binds strongly to an allosteric site. Kinetic studies using l-tyrosine as substrate indicated **2** and **3** competitively and non-competitively inhibit tyrosinase, respectively, which was supported by our docking results. In B16F10 cells, **3** significantly and concentration-dependently reduced α–MSH plus IBMX induced increases in cellular tyrosinase activity and melanin production and the similarity between these inhibitory patterns implied that the anti-melanogenic effect of **3** might be due to its tyrosinase-inhibitory ability. In addition, **2** and **3** exhibited strong antioxidant effects; for example, they reduced ROS and ONOO^–^ levels and exhibited radical scavenging activities, suggesting that these effects might underlie their anti-melanogenic effects. Furthermore, **3** suppressed the expressions of melanogenesis-associated proteins and genes in B16F10 cells. These results suggest (*Z*)-5-(substituted benzylidene)-3-cyclohexyl-2-thioxothiazolidin-4-one analogs offer a means of producing novel anti-melanogenesis agents.

## 1. Introduction

Aging is a time-dependent process of changes in the body’s biological functions [1], and ultraviolet (UV) radiation is the most important external cause of skin aging [2] and causes immune dysfunction and DNA damage, which substantially increase the risk of melanoma [2]. Melanogenesis results from a series of melanin-producing processes in melanosomes (organelles within melanocytes) and is the primary photoprotective response to acute and chronic UV-related skin damage [2]. On the other hand, abnormal melanin accumulation in certain skin regions causes aesthetic problems and pathological conditions such as lentigo, freckles, and melasma [3]. UV-induced melanogenesis in melanocytes is regulated by several molecular signaling pathways. For example, alpha-melanocyte-stimulating hormone (α–MSH) is the primary physiological inducer of melanogenesis. More specifically, the binding of α–MSH to melanocortin-1 receptors (MC1R) on the surfaces of melanocytes activates the protein kinase A (PKA) pathway, phosphorylates the cAMP response element-binding protein (CREB) transcription factor, and finally upregulates the expression of microphthalmia-associated transcription factor (MITF); a transcription factor for enzymes essential for melanogenesis (e.g., tyrosinase, tyrosinase-related protein 1 (TRP-1), and tyrosinase-related protein 2 (TRP-2)) [2,4].

Eumelanin and pheomelanin are the two main types of melanin. Eumelanin is responsible for the brown to black skin color, and pheomelanin for the yellow to red color, and both are synthesized from l-tyrosine by complex enzymatic and nonenzymatic processes [3,5]. l-Tyrosine is oxidized to l-dopa by the monophenolase activity of tyrosinase, and l-dopa is oxidized to dopaquinone (a common intermediate of eumelanin and pheomelanin production) by the diphenolase activity of tyrosinase. In the absence of thiols such as cysteine and glutathione, dopaquinone is transformed (by intramolecular Michael addition and subsequent oxidation) to dopachrome, which is finally converted to eumelanin by oxidation and polymerization. On the other hand, in the presence of thiols, dopaquinone is converted to pheomelanin by intermolecular Michael addition using the sulfhydryl groups of thiols, several complex oxidations, intramolecular cyclization, and decarboxylation reactions [5,6,7]. Dopachrome has been used as a biomarker for measuring the tyrosinase inhibitory activities of natural and synthetic compounds because it absorbs strongly at 475 nm [8,9,10,11,12,13] and its absorbance is reduced by tyrosinase inhibitors. Due to its rate-determining role in melanin synthesis, it is widely held that direct inhibition of tyrosinase and/or the downregulations of intracellular signaling pathways responsible for tyrosinase expression are attractive strategies for the treatment of hyperpigmentation disorders. A great number of compounds have been shown to inhibit melanin biosynthesis, as evidenced by in vitro experiments using mushroom tyrosinase and cultured pigment cells such as B16F10 cells [14,15,16,17,18,19]. However, few compounds are being used clinically as skin lightening agents because of safety issues and insufficient efficacy, and thus, a safe, effective, novel tyrosinase has yet to be realized [20,21,22].

Over the past decade, we have studied the tyrosinase-inhibitory activities of a considerable number of compounds containing the β-phenyl-α,β-unsaturated carbonyl (PUSC) scaffold (Figure 1), and shown that compounds of this type exhibit a proclivity to inhibit tyrosinase in vitro and in vivo [7,13,19,23,24,25,26,27,28]. In particular, cinnamamide derivatives containing the PUSC scaffold show potent tyrosinase inhibitory activity against mushroom and B16F10 cellular tyrosinase (Figure 1). The efficacies of anti-melanogenesis agents are closely related to their lipophilicities because melanocytes are present in the deepest epidermal layer, and thus, whitening agents must be easily absorbed by skin. For this reason, as shown in Figure 1, we cyclized cinnamamide derivatives using a dithionate (-C(=S)S-) group to introduce a 2-thioxothiazolidin-4-one (rhodanine) ring, which resulted in a log *P* increase of ~1. Eight (*Z*)-5-(substituted benzylidene)-3-cyclohexyl-2-thioxothiazolidin-4-one analogs with hydroxyl, methoxyl, and/or bromo substituents on the β-phenyl ring of the PUSC scaffold were designed, synthesized, and evaluated for mushroom tyrosinase and cellular tyrosinase inhibitory activities. In addition, we also investigated their anti-melanogenesis activities, antioxidant effects, and reactive nitrogen species (RNS) and reactive oxygen species (ROS) scavenging activities in B16F10 cells. Furthermore, the modes of action of tyrosinase inhibition by the analogs were determined using Lineweaver–Burk plots and in silico docking simulations, and their effects on melanogenesis-associated proteins and genes were explored.

## 2. Materials and Methods

### 2.1. General Methods

The eight (*Z*)-5-(substituted benzylidene)-3-cyclohexyl-2-thioxothiazolidin-4-ones were synthesized under nitrogen or argon. Anhydrous solvents were prepared by distillation over CaH_2_ or benzophenone/Na. Reactions were monitored by TLC (Merck TLC Silica gel 60F_245_ glass plates), and reaction mixtures were separated by recrystallization or flash column chromatography using silica gel (MP Silica 40-63, 60 Å). ^1^H and ^13^C NMR data were obtained using a JMTC-400/54/JJ/YH instrument (Japan Superconductor Technology, Kobe, Hyogo, Japan) at 400 and 100 MHz, respectively, and 500 MHz ^1^H NMR and 125 MHz ^13^C NMR were obtained using a Varian Unity AS500 unit (Agilent Technologies, Santa Clara, CA, USA). Chemical shifts are displayed in ppm (parts per million), and residual solvent peaks were used as reference peaks (DMSO-*d*_6_: *δ*_H_ 2.48 and *δ*_C_ 39.9 and CDCl_3_: *δ*_H_ 7.27 and *δ*_C_ 77.0). Proton coupling constants (*J*) are recorded in hertz (Hz), and the splitting patterns are described as: m (multiplet), s (singlet), brs (broad singlet), d (doublet), brt (broad triplet), dd (double to doublets), and tt (triplet of triplets). All reagents were purchased from Daejung Chemical & Materials Co. Ltd. (Gyeonggi-do, Korea), SEJIN CI Co. (Seoul, Korea), Sigma-Aldrich (St. Louis, MO, USA), or ThermoFisher Scientific (Seoul, Korea).

#### 2.1.1. Synthesis of *N*-Cyclohexylrhodanine (9)

Trimethylamine (4.91 mL, 35.23 mmol) was added to a stirred solution of methyl 2-mercaptoacetate (3.15 mL, 35.23 mmol) and cyclohexyl isothiocyanate (5.00 mL, 35.26 mmol) in dichloromethane (80 mL) and stirred for 12 h at room temperature (RT). After evaporation of volatiles, the resultant residue was recrystallized from a mixture of dichloromethane, hexane, and diethyl ether to give *N*-cyclohexylrhodanine (**9**, 6.26 g, 82.6%) as a solid.

^1^H NMR (500 MHz, CDCl_3_) *δ* 4.85 (tt, 1H, *J* = 12.3, 3.5 Hz, 1-H), 3.81 (s, 2H, COCH_2_), 2.33–2.26 (m, 2H, cyclohexyl), 1.88–1.82 (m, 2H, cyclohexyl), 1.69–1.60 (m, 3H, cyclohexyl), 1.39–1.30 (m, 2H, cyclohexyl), 1.27–1.18 (m, 1H, cyclohexyl); ^13^C NMR (125 MHz, CDCl_3_) *δ* 202.4, 174.3, 58.4, 33.9, 27.5, 26.0, 25.0.

#### 2.1.2. General Procedure for the Syntheses of (Z)-5-(Substituted benzylidene)-3-cyclohexyl-2-thioxothiazolidin-4-one Analogues **1**–**8**

A solution of **9** (100 mg, 0.46 mmol) and a substituted benzaldehyde (1.0 equiv. for aldehydes **g**, and **h**; 1.1 equiv. for aldehydes **a**–**f**) in acetic acid (1.0 mL) was refluxed in the presence of NaOAc (257 mg, 3.13 mmol) for 5–20 h. After cooling, water was added to the reaction mixture, and the resulting solid was filtered and washed with water, dichloromethane, methanol, and/or a hexane and dichloromethane mixture (3:1) to give **1**–**8** as solids in yields of 65–91%.

##### (*Z*)-3-Cyclohexyl-5-(4-hydroxybenzylidene)-2-thioxothiazolidin-4-one (**1**)

^1^H NMR (500 MHz, DMSO-*d*_6_) *δ* 10.46 (s, 1H, OH), 7.61 (s, 1H, vinylic H), 7.47 (d, 2H, *J* = 9.0 Hz, 2-H, 6-H), 6.92 (d, 2H, *J* = 9.0 Hz, 3-H, 5-H), 4.89 (brt, 1H, *J* = 12.0 Hz, 1′-H), 2.34–2.28 (m, 2H, cyclohexyl), 1.83–1.71 (m, 2H, cyclohexyl), 1.67–1.63 (m, 3H, cyclohexyl), 1.33–1.25 (m, 2H, cyclohexyl), 1.20–1.12 (m, 1H, cyclohexyl); ^13^C NMR (100 MHz, DMSO-*d*_6_) *δ* 194.3, 167.8, 161.0, 133.8, 133.7, 124.6, 117.3, 117.1, 57.9, 27.9, 26.1, 25.3.

##### (*Z*)-3-Cyclohexyl-5-(3,4-dihydroxybenzylidene)-2-thioxothiazolidin-4-one (**2**)

^1^H NMR (500 MHz, DMSO-*d*_6_) *δ* 9.99 (s, 1H, OH), 9.54 (s, 1H, OH), 7.53 (s, 1H, vinylic H), 7.02–6.99 (m, 2H, 2-H, 6-H), 6.88 (d, 1H, *J* = 8.5 Hz, 5-H), 4.89 (brt, 1H, *J* = 12.0 Hz, 1′-H), 2.35–2.28 (m, 2H, cyclohexyl), 1.84–1.80 (m, 2H, cyclohexyl), 1.67–1.63 (m, 3H, cyclohexyl), 1.33–1.25 (m, 2H, cyclohexyl), 1.20–1.12 (m, 1H, cyclohexyl); ^13^C NMR (100 MHz, DMSO-*d*_6_) *δ* 194.4, 167.7, 149.9, 146.6, 134.1, 125.7, 125.0, 117.2, 117.1, 117.0, 57.9, 27.9, 26.1, 25.4.

##### (*Z*)-3-Cyclohexyl-5-(2,4-dihydroxybenzylidene)-2-thioxothiazolidin-4-one (**3**)

^1^H NMR (500 MHz, DMSO-*d*_6_) *δ* 10.66 (s, 1H, OH), 10.33 (s, 1H, OH), 7.84 (s, 1H, vinylic H), 7.15 (d, 1H, *J* = 9.0 Hz, 6-H), 6.42–6.38 (m, 2H, 2-H, 5-H), 4.88 (brt, 1H, *J* = 12.0 Hz, 1′-H), 2.35–2.27 (m, 2H, cyclohexyl), 1.82–1.78 (m, 2H, cyclohexyl), 1.64–1.60 (m, 3H, cyclohexyl), 1.32–1.23 (m, 2H, cyclohexyl), 1.18–1.11 (m, 1H, cyclohexyl); ^13^C NMR (125 MHz, DMSO-*d*_6_) *δ* 194.4, 167.8, 162.9, 160.2, 131.8, 129.3, 114.9, 112.6, 109.2, 102.8, 57.6, 27.7, 25.9, 25.1.

##### (*Z*)-3-Cyclohexyl-5-(4-hydroxy-3-methoxybenzylidene)-2-thioxothiazolidin-4-one (**4**)

^1^H NMR (500 MHz, CDCl_3_) *δ* 7.55 (s, 1H, vinylic H), 7.07 (dd, 1H, *J* = 8.5, 2.0 Hz, 6-H), 7.00 (d, 1H, *J* = 8.5 Hz, 5-H), 6.95 (d, 1H, *J* = 2.0 Hz, 2-H), 6.01 (s, 1H, 4-OH), 5.01 (brt, 1H, *J* = 12.0 Hz, 1′-H), 3.96 (s, 3H, OCH_3_), 2.49–2.38 (m, 2H, cyclohexyl), 1.91–1.86 (m, 2H, cyclohexyl), 1.72–1.68 (m, 3H, cyclohexyl), 1.43–1.35 (m, 2H, cyclohexyl), 1.30–1.26 (m, 1H, cyclohexyl); ^13^C NMR (100 MHz, CDCl_3_) *δ* 194.1, 168.2, 148.4, 147.1, 132.8, 126.2, 126.2, 119.3, 115.4, 111.9, 58.2, 56.1, 27.9, 26.2, 25.1.

##### (*Z*)-3-Cyclohexyl-5-(3-hydroxy-4-methoxybenzylidene)-2-thioxothiazolidin-4-one (**5**)

^1^H NMR (500 MHz, CDCl_3_) *δ* 7.52 (s, 1H, vinylic H), 7.05 (d, 1H, *J* = 2.0 Hz, 2-H), 7.03 (dd, 1H, *J* = 8.5, 2.0 Hz, 6-H), 6.92 (d, 1H, *J* = 8.5 Hz, 5-H), 5.74 (brs, 1H, 3-OH), 5.01 (brt, 1H, *J* = 12.0 Hz, 1′-H), 3.95 (s, 3H, OCH_3_), 2.48–2.36 (m, 2H, cyclohexyl), 1.92–1.85 (m, 2H, cyclohexyl), 1.73–1.66 (m, 3H, cyclohexyl), 1.42–1.34 (m, 2H, cyclohexyl), 1.29–1.23 (m, 1H, cyclohexyl); ^13^C NMR (125 MHz, CDCl_3_) *δ* 194.3, 168.1, 148.7, 146.1, 132.3, 127.1, 124.3, 120.1, 116.0, 111.0, 58.0, 56.1, 27.8, 26.1, 25.1.

##### (*Z*)-3-Cyclohexyl-5-(4-hydroxy-3,5-dimethoxybenzylidene)-2-thioxothiazolidin-4-one (**6**)

^1^H NMR (500 MHz, CDCl_3_) *δ* 7.51 (s, 1H, vinylic H), 6.70 (s, 2H, 2-H, 6-H), 5.93 (s, 1H, 4-OH), 5.00 (brt, 1H, *J* = 12.0 Hz, 1′-H), 3.94 (s, 6H, 2 × CH_3_), 2.48–2.37 (m, 2H, cyclohexyl), 1.92–1.85 (m, 2H, cyclohexyl), 1.73–1.66 (m, 3H, cyclohexyl), 1.42–1.34 (m, 2H, cyclohexyl), 1.32–1.25 (m, 1H, cyclohexyl); HH^13^C NMR (125 MHz, CDCl_3_) *δ* 193.9, 167.3, 147.5, 137.6, 132.8, 125.1, 119.7, 107.7, 58.1, 56.4, 27.8, 26.1, 25.0.

##### (*Z*)-5-(3-Bromo-4-hydroxybenzylidene)-3-cyclohexyl-2-thioxothiazolidin-4-one (**7**)

^1^H NMR (500 MHz, CDCl_3_) *δ* 7.61 (d, 1H, *J* = 2.0 Hz, 2-H), 7.48 (s, 1H, vinylic H), 7.36 (dd, 1H, *J* = 8.5, 2.0 Hz, 6-H), 7.11 (d, 1H, *J* = 8.5 Hz, 5-H), 5.94 (s, 1H, 4-OH), 5.00 (brt, 1H, *J* = 12.0 Hz, 1′-H), 2.47–2.36 (m, 2H, cyclohexyl), 1.92–1.85 (m, 2H, cyclohexyl), 1.73–1.66 (m, 3H, cyclohexyl), 1.42–1.35 (m, 2H, cyclohexyl), 1.32–1.24 (m, 1H, cyclohexyl); ^13^C NMR (125 MHz, CDCl_3_) *δ* 193.6, 167.9, 154.1, 134.3, 131.7, 130.1, 127.8, 121.2, 116.9, 111.3, 58.1, 27.8, 26.0, 25.0.

##### (*Z*)-3-Cyclohexyl-5-(3,5-dibromo-4-hydroxybenzylidene)-2-thioxothiazolidin-4-one (**8**)

^1^H NMR (400 MHz, DMSO-*d*_6_) *δ* 10.87 (brs, 1H, 4-OH), 7.75 (s, 2H, 2-H, 6-H), 7.59 (s, 1H, vinylic H), 4.85 (brt, 1H, *J* = 12.0 Hz, 1′-H), 2.32–2.18 (m, 2H, cyclohexyl), 1.82–1.75 (m, 2H, cyclohexyl), 1.66–1.56 (m, 3H, cyclohexyl), 1.31–1.22 (m, 2H, cyclohexyl), 1.19–1.09 (m, 1H, cyclohexyl); ^13^C NMR (125 MHz, DMSO-*d*_6_) *δ* 193.7, 167.3, 153.5, 134.7, 130.3, 127.9, 121.2, 112.9, 57.9, 27.8, 26.0.

### 2.2. Mushroom Tyrosinase Activity Assay

The mushroom tyrosinase inhibitory activities of analogs **1**–**8** were measured as described previously with minor modifications [13]. Kojic acid and l-tyrosine were used as a positive reference material and a substrate for the assay, respectively. In brief, 20 µL of an aqueous mushroom tyrosinase solution (1000 units/mL) and 170 µL of a substrate mixture containing 345 µM of l-tyrosine and 17.2 mM of sodium phosphate buffer (pH 6.5) were added to wells of a 96-well microplate. Analogs or kojic acid at 3–5 different concentrations (10 µL, DMSO solution) were then added to wells and incubated for 30 min at 37 °C. Amounts of dopachrome produced were assessed by measuring optical densities at 475 nm using a microplate reader (VersaMax^®^, Molecular Devices, Sunnyvale, CA, USA). IC_50_ values of analogs and kojic acid were determined using the results obtained. Experiments were independently conducted three times.

### 2.3. In Silico Docking Simulations of Analogues ***2**, **3***, and Kojic Acid with Mushroom Tyrosinase Using Schrödinger Suite Software

In silico docking simulations of analogues **2**, **3**, or kojic acid with mushroom tyrosinase were carried out using Schrödinger Suite (2021–2) with minor modifications, as previously described [3]. The mushroom tyrosinase crystal structure (*Agaricus bisporus*, PDB ID: 2Y9X) was obtained from the Protein Data Bank (PDB) and transferred to Protein Preparation Wizard in Maestro 12.4. The mushroom tyrosinase crystal structure was processed, and unnecessary protein chains were deleted. To optimize the crystal structure, water molecules > 3 Å from the enzyme were deleted, and hydrogen atoms were added. The ligand (tropolone) bound to the active site of the enzyme crystal structure obtained from the PDB was removed, and the space where the ligand was bound was allocated as a glide grid was inserted in the space created [29,30,31]. The structures of **2**, **3**, and kojic acid were imported into the entry list of Maestro in CDXML format and prepared using LigPrep before docking simulation. Ligands were then docked to the tyrosinase glide grid [32]. Binding affinities of ligands and interactions between ligands and mushroom tyrosinase were obtained using the extra precision (XP) glide method [33].

### 2.4. In Silico Docking Simulations of Analog ***3*** and Cinnamic Acid with Mushroom Tyrosinase Using AutoDock Vina Software

In silico docking simulations between analog **3** and mushroom tyrosinase were performed to identify the allosteric binding site in tyrosinase, as previously described with minor modifications [34]. Cinnamic acid was used as a reference compound because it was reported that cinnamic acid binds to the allosteric site of mushroom tyrosinase [35]. The 3D structures of **3** and cinnamic acid were created using Chem3D Pro 12.0 software, and the 3D crystal structure of tyrosinase (*Agaricus Bisporus*) was obtained from the PDB (PDB ID: 2Y9X). The 3D structures of analog **3** and cinnamic acid were docked on the entire tyrosinase structure using AutoDock Vina 1.2.0 (The Scripps Research Institute, La Jolla, CA, USA). and Chimera 1.15 (UCSF, San Francisco, CA, USA), and their binding affinities were calculated. LigandScout ver. 4.3.0 (Inte:Ligand GmbH, Vienna, Austria) was used to create a pharmacophore model indicating possible interactions between analog **3** and amino acid residues of tyrosinase.

### 2.5. In Silico Docking Simulations of Analogues ***2*** and ***3*** and Kojic Acid with a Human Tyrosinase Homology Model

The homology model for human tyrosinase was generated using Schrödinger Suite (2020–2) (NY, USA) and the SWISS-MODEL online server (Moudon, Switzerland) (https://swissmodel.expasy.org/interactive). The protein sequence of human tyrosinase (P14679) was obtained from the UniProt database, and the homology model was created using the SWISS-MODEL online server using a tyrosinase-related protein-1 (PDB ID: 5M8Q, TRP-1) template containing zinc atoms at the active site. The homology model structure was further processed using Schrödinger Suite and validated using Schrödinger prime (a homology modeling tool in Schrödinger Suite). Analogs **2** and **3** and kojic acid were docked with the human tyrosinase model using the protocols described above for mushroom tyrosinase docking simulation.

### 2.6. Kinetic Studies on Mushroom Tyrosinase Inhibition by Analogues ***2*** and ***3***

Lineweaver–Burk plots of analogues **2** and **3** were used to determine modes of mushroom tyrosinase inhibition in the presence of l-tyrosine as substrate. Briefly, 20 µL of an aqueous mushroom tyrosinase solution (200 units/mL), and 170 µL of an aqueous solution consisting of l-tyrosine at various concentrations (final concentrations: 0, 1, 2, 4, 8, or 16 mM), and sodium phosphate buffer (pH 6.5, final concentration: 14.7 mM) were added to the wells of a 96-well microplate. Subsequently, analogues **2** and **3** at four different concentrations (10 µL; final concentrations 0, 5, 10, or 20 µM in DMSO solution for **2**, and concentrations of 0, 1, 2, or 4 µM in DMSO solution for **3**) were added to wells. Initial rates of dopachrome formation were determined by measuring changes in absorbance at 475 nm per min (ΔOD_475_/min) using a microplate reader (VersaMax^®^, Molecular Devices, Sunnyvale, CA, USA). Maximum velocities (V_max_) were calculated from Lineweaver–Burk plots obtained using six different l-tyrosine concentrations. The modes of mushroom tyrosinase inhibition by the analogs were determined using the locations of plot convergence points.

### 2.7. Cell Culture

Murine melanoma B16F10 cells were purchased from the American Type Culture Collection (ATCC, Manassas, VA, USA). Fetal bovine serum (FBS), trypsin, phosphate buffer solution (PBS), penicillin, streptomycin, and Dulbecco’s modified Eagle’s medium (DMEM) were purchased from Gibco (Grand Island, NY, USA). B16F10 cells were cultured in a DMEM solution containing heat-inactivated 10% FBS, penicillin (100 IU/mL), and streptomycin (100 µg/mL) in a humidified 5% CO_2_ atmosphere at 37 °C. Anti-tyrosinase activity, anti-melanogenesis activity, and cell viability assays were conducted on cultured cells in 96- or 6-well plates.

### 2.8. Viabilities of B16F10 Murine Melanoma Cells Treated with Analogues ***2*** or ***3***

The cytotoxic effects of analogs **2** and **3** were examined using B16F10 murine melanoma cells, as previously described [4]. Cell viabilities were determined using the EZ-Cytox cell viability assay kit (Daeil Lab Service Co. Ltd., Seoul, Korea). Briefly, B16F10 cells (density 1 × 10^4^ cells/well) were added to the wells of a 96-well microplate and cultured in a humidified 5% CO_2_ atmosphere at 37 °C for 24 h. Cells were then exposed to analogs **2** or **3** at 0, 1, 2, 5, 10, or 20 µM and incubated in a humidified 5% CO_2_ atmosphere for 24, 48, or 72 h at 37 °C. At these time points, cells were exposed to 10 µL of EZ-Cytox solution and incubated for 2 h at 37 °C. Optical densities of wells were then measured at 450 nm using a microplate reader (VersaMax^®^, Molecular Devices, Sunnyvale, CA, USA) to assess cell viability. Experiments were conducted independently 3 times.

### 2.9. Cellular Tyrosinase Activity Assay

The cellular tyrosinase activity of analog **3** was studied by measuring the rate of l-dopa oxidation, as previously described with slight modification [11]. Kojic acid was used as the positive reference control. Briefly, cultured B16F10 cells were seeded in 6-well culture plate at 1 × 10^4^ cells/well and incubated for 12 h under cell culture conditions to allow them to adhere to well bottoms. Cells were exposed to analog **3** at four different concentrations (0, 5, 10, or 20 µM), or kojic acid (20 µM) for 1 h and then co-treated with IBMX (Sigma-Aldrich, St. Louis, MO, USA) (200 µM) and α–MSH (Sigma-Aldrich, St. Louis, MO, USA) (1 µM) for 72 h to stimulate cellular tyrosinase activity. Cells were then washed with PBS twice and exposed to a lysis buffer solution (100 µL) comprised of 5 µL of 20% Triton X-100, 5 µL of 2 mM PMSF, and 90 µL of 50 mM phosphate buffer in pH 6.5. Lysates were kept at −80 °C for 30 min, defrosted, and centrifuged at 12,000 rpm for 30 min at 4 °C. Supernatants (80 µL) were mixed with 20 µL of 2 mg/mL of l-dopa in a 96-well plate, and absorbance of each well was measured at 475 nm every 10 min for 1 h at 37 °C using a microplate reader (VersaMax^®^, Molecular Devices, Sunnyvale, CA, USA). Experiments were performed independently 3 times.

### 2.10. Melanin Content Analysis

Melanin content assays were performed to determine the effects of analog **3** on cellular melanin formation, as previously described with minor modification [36]. Kojic acid was used as a positive reference control. Briefly, cultured B16F10 cells were seeded in 6-well culture plates at 1 × 10^4^ cells/well and incubated for 12 h under culture conditions to allow them to adhere to well bottoms. Cells were then exposed to analog **3** at four concentrations (0, 5, 10, or 20 µM) or kojic acid (20 µM) for 1 h and then co-treated with IBMX (200 µM) and α–MSH (1 µM) for 72 h under culture conditions. Cells were then washed twice with PBS and detached from wells by incubation with Trypsin/EDTA. After centrifugation, detached cell pellets were dissolved in 1N–NaOH solution (100 µL), incubated for 1 h at 60 °C, and mixed to solubilize melanin released by cells. The ability of analog **3** to inhibit melanin production was determined by measuring well optical densities (ODs) at 405 nm using a microplate reader (VersaMax^®^, Molecular Devices, Sunnyvale, CA, USA). Melanin contents were calculated by dividing the ODs of samples by the control OD. Experiments were conducted independently 3 times.

### 2.11. DPPH Radical Scavenging Activity Assay

The DPPH radical scavenging activities of analogs **1**–**8** were investigated as previously described with minor modification [19]. l-Ascorbic acid was used as a positive control. In brief, 20 µL of analogs (5 mM) in DMSO or 5 mM l-ascorbic acid in distilled water were mixed with 180 µL of 0.2 mM DPPH in methanol in a 96-well microplate. After the microplate had been stored in the dark for 30 min, optical densities (ODs) were measured at 517 nm using a microplate reader (VersaMax^®^, Molecular Devices, Sunnyvale, CA, USA). DPPH radical scavenging activities were determined using the following formula:DPPH radical scavenging activity (%) = 100 × (Ac − As)/Ac(1)
where Ac indicates the optical density of the non-treated control and As indicates the optical density of test sample. Experiments were conducted independently 3 times.

### 2.12. ABTS Cation Radical Scavenging Activity Assay

The free radical scavenging activities of analogs **1**–**8** and Trolox were determined using an ABTS radical cation decolorization assay (SEJIN CI Co., Seoul, Korea), as previously described with minor modification [37]. Trolox was used as the positive control. ABTS radicals were produced by adding 10 mL of 7 mM ABTS in distilled water to 10 mL of 2.45 mM aqueous potassium persulfate solution. The mixture was kept in the dark for 12–16 h at RT until the reaction was complete and absorbance had stabilized. The prepared free radical solution was then diluted with methyl alcohol so that the optical density at 734 nm was 0.70 ± 0.02. Test samples (10 µL, final concentration at 100 µM) dissolved in EtOH/DMSO (9:1 *v*/*v*) were added to the diluted free radical solution (90 µL) in a 96-well plate. After standing at RT for 2 min in the dark, well absorbances were measured at 734 nm, and ABTS radical scavenging activities were calculated using the following formula:ABTS free radical scavenging activity (%) = (Ac − As) × 100/Ac(2)
where Ac indicates the absorbance of the non-treated control, and As indicates the absorbances of test samples. Experiments were conducted independently 3 times.

### 2.13. ROS Scavenging Activity Assay

The ROS scavenging activities of analogs **1**–**8** and Troloxwere determined as previously described by Ali et al. [38] and Lebel and Bondy [39]. Briefly, 2.5 mM DCFH-DA (2’,7’-dichlorodihydrofluorescein diacetate) (Molecular Probes, Eugene, OR, USA) solution was mixed with esterase (1.5 units/mL) in 50 mM phosphate buffer (pH 7.4), held for 30 min at 37 °C, and keep on ice in the dark until required. Tested compounds (10 µL; final concentrations: 5, 10, and 20 µM) dissolved in DMSO and SIN-1 (3-morpholinosydnonimine) (10 µM) and 180 µL of phosphate buffer in the wells of a black 96-well plate for 30 min. The 50 µL of DCFH-DA-esterase mixture (20 µM) was added. Fluorescence intensities of oxidized DCFH (DCF or 2′,7′-dichlorofluorescein) were measured using a microplate reader (Berthold Advances GmbH & Co., Bad Wildbad, Germany) with excitation and emission wavelengths of 485 and 530 nm, respectively. Trolox was used as the reference control.

### 2.14. Peroxynitrite (ONOO^−^) Scavenging Activity Assay

Peroxynitrite-scavenging activities were determined as described by Kooy et al. [40] with modification. This method monitors the fluorescence of rhodamine 123, which is formed rapidly when non-fluorescent DHR 123 (dihydrorhodamine 123) (Molecular Probes, Eugene, OR, USA) in the presence of peroxynitrite. The rhodamine buffer solution (pH 7.4) included of 50 mM sodium phosphate, 90 mM sodium chloride, 5 mM potassium chloride, and 100 µM DTPA. The final concentration of DHR 123 was 5 µM. In this assay, the buffer solution was prepared prior to use and then stood on ice. Test compounds (10 µL, final concentrations: 5, 10, and 20 µM) dissolved in DMSO and added to 10 µL SIN-1 (10 µM) and 180 µL rhodamine buffer solution in the wells of a black 96-well plate. Fluorescence intensities of oxidized DHR 123 were measured using a fluorescence plate reader (Berthold Advances GmbH & Co., Bad Wildbad, Germany) with an excitation and emission wavelengths of 485 and 535 nm, respectively. Peroxynitrite scavenging activity was calculated after subtracting background fluorescence from final fluorescence intensity. Results are expressed as means ± standard deviation (SD), and l-penicillamine was used as a positive control.

### 2.15. Extraction of Cytosolic and Nuclear Proteins from Cells

Western blotting was used to determine tyrosinase and MITF protein levels. Briefly, B16F10 cells were seeded in 60 mm culture dishes in the presence or absence of α–MSH (1 µM) plus IBMX (200 µM) with or without analog **3** (5, 10, and 20 µM for final concentrations) for 24 h, washed twice with ice-cold PBS and centrifuged for 10 min at 12,000× *g*. Cell pellets were suspended in 10 mM Tris (pH 8.0) containing 0.1% NP-40, 1 mM DTT, 1.5 mM MgCl_2_, 1 mM PMSF, and protease inhibitors (GenDEPOT, Barker, TX, USA), incubated on ice for 15 min, and centrifuged for 10 min at 12,000× *g* at 4 °C. Supernatants were used as cytosolic fractions. Pellets were resuspended in 10 mM Tris (pH 8.0) containing 1 mM PMSF, 100 mM NaCl, 50 mM KCl, and protease inhibitor incubated on ice for 15 min, and centrifuged at 12,000× *g* for 10 min at 4 °C. The supernatants contain nuclear fractions. The protein concentrations were determined using a bicinchoninic acid (BCA)™ assay kit (Pierce, Rockford, IL, USA).

### 2.16. Western Blot Analysis

Lysed protein samples from cells were boiled for 10 min in sample-loading buffer (0.1% bromophenol blue, 25% glycerol, 60 mM Tris-HCl (pH 6.8), 2% SDS, and 0.5% β-mercaptoethanol) at a volume ratio of 1:4. Proteins in lysed cytosolic and nuclear fractions were separated by SDS-PAGE using 9% acrylamide gels and then transferred to polyvinylidene fluoride (PVDF) membranes (Millipore, Burlington, MA, USA) at 25 V for 10 min using in a semi-dry transfer system (Bio-Rad Laboratories, Hercules, CA, USA). After transfer, membranes were instantly placed in blocking buffer in 5% dry non-fat milk in TBST (10 mM Tris, 100 mM NaCl, 0.1% Tween-20 at pH 7.5) for 1 h at RT, incubated overnight with appropriate primary antibodies (1:500 to 1:2000 dilution) at 4 °C, washed with TBST buffer (3 × 10 min), and incubated with horseradish peroxidase-conjugated secondary antibodies (anti-mouse or anti-goat antibody (dilution 1:5000)) for 1 h at RT. Antibodies specific for MITF (dilution 1:500), TFIIB (dilution 1:2000), β-actin (dilution 1:2000), and tyrosinase (dilution 1:1000) were obtained from Santa Cruz Biotechnology (Santa Cruz, CA, USA). Immunoreactive proteins were detected using the enhanced chemiluminescence (ECL) detection reagent in the SuperSignal^®^ West Pico Chemiluminescent Substrate kit (Advansta, San Jose, CA, USA). Western blot bands were visualized using a Davinch-Chemi imager™ (Davinch-K, Seoul, Korea), and quantification was conducted by densitometric analysis using CS analyzer software ver. 3.00.1010 (VS-Tannheim, Germany) [13].

### 2.17. RNA Extraction and Quantitative Real Time-Polymerase Chain Reaction (RT-qPCR)

B16F10 cells were extracted using GeneAll RiboEx Total RNA extraction reagent (GeneAll Biotechnology, Seoul, Korea) and transferred to tubes, added with chloroform (0.1 mL), shaken vigorously for 30 sec, and centrifuged for 15 min at 12,000× *g*. Aqueous phases were transferred to fresh tubes, and equal volumes of isopropanol were added. Samples were then incubated at 4 °C for 10 min and centrifuged for 15 min at 12,000× *g* at 4 °C. Supernatants were discarded, and RNA pellets were washed once with 0.8 mL of 70% ethanol and by vortexing briefly and centrifuged for 5 min at 7500× *g* at 4 °C. Pellets were dried at 56 °C for 10–15 min and dissolved in 30 µL of diethyl pyrocarbonate (DEPC)-treated water. The levels of RNA were determined by a Nanodrop spectrophotometer (NANO-200, Allsheng, Hangzhou, China). Complementary DNA (cDNA) was synthesized from total RNA (2 µg) using SuPrime Script RT Premix (with random hexamer) cDNA Synthesis Kit (GeNet Bio, Daejeon, Korea). RT-qPCR amplification was performed using AccuPower^®^ 2X Green Star^TM^ qPCR Mater Mix (Bioneer, Daejeon, Korea) and the CFX Connect^TM^ Real-time PCR Detection System (Bio-Rad Laboratories, Hercules, CA, USA). RT-qPCR primers specific for *tyrosinase*, *TRP-1*, and *GAPDH* were purchased from Bioneer Inc. (Daejeon, Korea). The expression levels of the target genes were normalized to GAPDH [41]. The primer sequences used are listed in Table 1.

### 2.18. Statistical Analysis

The significance of differences between groups was determined by one-way analysis of variance (ANOVA) followed by Newman–Keuls test. The analysis was performed using GraphPad Prism 5 software (La Jolla, CA, USA). Results are displayed as means ± SEMs, and two-sided *p*-values of <0.05 were considered significant.

## 3. Results and Discussion

### 3.1. Chemistry

According to our accumulated data, compounds with the PUSC scaffold and hydroxyl substituents on the β-phenyl group often exhibit potent tyrosinase inhibitory activity [3,7,13,42,43]. In the present study, we designed and synthesized a series of eight (*Z*)-5-(substituted benzylidene)-3-cyclohexyl-2-thioxothiazolidin-4-one analogs with at least one hydroxyl on the β-phenyl ring of the PUSC entity. We considered *N*-cyclohexylrhodanine (**9**) an appropriate intermediate for the preparation of analogs **1**–**8**. Intermediate **9** [44,45] was prepared by coupling methyl 2-mercaptoacetate and *N*-cyclohexyl isothiocyanate in the presence of triethylamine in 83% yield (Figure 1). The synthesis of (*Z*)-5-(substituted benzylidene)-3-cyclohexyl-2-thioxothiazolidin-4-one analogs was successfully achieved under Knoevenagel condensation conditions. Condensation of **9** with eight benzaldehydes **a**–**h** with hydroxyl, methoxyl, and/or bromo substituents on the phenyl ring in the presence of NaOAc in acetic acid produced analogs **1**–**8** in yields of 65–91% (Table 1) [46]. The structures of the final compounds were determined by ^1^H and ^13^C NMR spectroscopy. The trisubstituted exocyclic methylene geometry of analogs **1**–**8** were determined using ^3^*J*_C4-Hβ_ values of C4 measured in proton-coupled ^13^C mode [3,47,48]. When the olefinic H_β_-atom and the carbonyl group are on the same side, ^3^*J*_C4-Hβ_ values lie in the range 3.6–7.0 Hz, and when on the opposite side, values are typically ≥ 10 Hz [47]. The ^13^C NMR spectrum of analog **3** was obtained in proton-coupled mode, and the C4 triplet showed that ^3^*J*_C4-Hβ_ and ^3^*J*_C4-H1__′_ coupling constants were the same at 5.4 Hz (Appendix A). Furthermore, a ^3^*J*_C4-Hβ_ value of 5.4 Hz indicated that the tri-substituted exocyclic methylene had a (*Z*)-configuration, which was supported by the H_β_ chemical shifts of analogs **1**–**8** in ^1^H NMR spectra. Chemical shifts of H_β_ ranged from 7.48 to 7.83 ppm, and we attributed this deshielding to a cis positioned H_β_ adjacent to the carbonyl group of the rhodanine ring, due to the anisotropic effect of the carbonyl. Although *E* and *Z* geometrical isomers were possible during Knoevenagel condensations, only *Z*-isomers were obtained, suggesting that this configuration has greater thermodynamic stability due to intramolecular hydrogen bonding between H_β_ and the carbonyl oxygen (Figure 1) and less steric hindrance between the carbonyl group and the β-phenyl ring [49].

### 3.2. Mushroom Tyrosinase Inhibition

Inhibitory activities of the eight analogs and kojic acid against mushroom tyrosinase were assayed using l-tyrosine as a substrate, as described previously [23]. Table 2 shows the IC_50_ values of the analogs. Kojic acid is typically used as a positive control for tyrosinase inhibition evaluations, and in this study, potently inhibited tyrosinase (IC_50_ 25.26 ± 1.10 µM). Although analog **1** with a 4-hydroxyl substituent on the β-phenyl ring had less mushroom tyrosinase inhibitory activity than kojic acid, it effectively inhibited tyrosinase (IC_50_ 65.39 ± 7.82 µM). Insertion of a methoxyl into position 3 of the β-phenyl ring in **1** greatly reduced inhibitory activity (IC_50_ of analog **4** was >200 µM). Interestingly, the exchange of substituents on the β-phenyl ring of **4** restored inhibitory activity (IC_50_ value of analog **5** was 54.98 ± 11.91 µM). This result is consistent with previous studies [50,51,52,53] that reported compounds with 3-hydroxy-4-methoxyl substituents on the β-phenyl ring of the PUSC scaffold usually exhibit moderate to potent tyrosinase inhibitory activity, while compounds with a 4-hydroxy-3-methoxyl substituent on the β-phenyl ring exhibit weak or no tyrosinase inhibitory activity. The introduction of an additional bromo substituent at position 3 of the β-phenyl ring in **1** did not influence inhibitory activity (IC_50_ of analog **7**: 66.23 ± 9.44 µM), but the introduction of two bromo substituents into positions 3 and 5 of the β-phenyl ring of **1** greatly reduced inhibitory activity (IC_50_ value of analog **8**: >200 µM). On the other hand, insertion of an additional hydroxyl substituent into positions 2 or 3 of the β-phenyl ring of **1** enhanced tyrosinase inhibitory activity by 63- and 13-fold, respectively, which showed analog **2** (IC_50_ value = 5.21 ± 0.86 µM) with a 3,4-dihydroxyphenyl and analog **3** (IC_50_ value = 1.03 ± 0.14 µM) with a 2,4-dihydroxyphenyl inhibited tyrosinase 5- and 25-fold more than kojic acid, respectively. Since analogs **2** and **3** potently inhibited mushroom tyrosinase in the presence of l-tyrosine as substrate, we examined their abilities to inhibit mushroom tyrosinase in the presence of l-dopa (another tyrosinase substrate). The IC_50_ value of **2** (39.66 ± 3.43 µM) was similar to that of kojic acid (38.60 ± 1.41 µM) in the presence of l-dopa, and that of **3** (IC_50_ = 10.55 ± 2.92 µM) more potently inhibited tyrosinase inhibition than **2** or kojic acid. These results show the tyrosinase inhibitory activities of these analogs are sensitively dependent on the number and positions of substituents on the β-phenyl ring of the PUSC scaffold. In particular, they show that analogs with a resorcinol (2,4-dihydroxyphenyl) or catechol (3,4-dihydroxyphenyl) moiety exhibit potent tyrosinase inhibitory potency.

Since the efficacy of anti-melanogenesis agents is closely associated with skin absorbance, we examined the log *p* (partition coefficient) values of analogs **1**–**8** using ChemDraw Ultra Ver. 12.0. Log *p* values are summarized in Table 2. All analogs had higher log *p* values (3.04 to 5.09) than kojic acid (log *P* = −2.45) and the corresponding cinnamamides (log *P* = 2.29–4.33) (Figure 1), indicating that the partition coefficients of analogs were 10^5^–10^7^-fold greater than that of kojic acid.

### 3.3. In Silico Docking Simulation between Tyrosinase and Analogues ***2*** and ***3***

#### 3.3.1. Docking Simulations of Analogues **2** and **3** with Mushroom Tyrosinase

In silico docking simulation studies of **2** and **3** were conducted using the crystal structure of mushroom (*Agaricus bisporus*) tyrosinase (PDB ID: 2Y9X) and Schrodinger Suite (release 2021-1). Kojic acid was used as a positive control. Analogs **2** and **3** and kojic acid were docked at the active site after removing the tropolone from the tyrosinase active site in the 3D crystal structure of mushroom tyrosinase. Figure 2 shows interactions between mushroom tyrosinase and **2**, **3**, and kojic acid in 2D and 3D.

Kojic acid interacted with the active site of tyrosinase by coordinating with Cu401 using its hydroxymethyl, π–π stacking with His263 using its 4-pyranone ring, and hydrogen bonding with Met280 using its 5-hydroxyl. Analog **2** formed two π–π stacking interactions with His259 and His263 using the β-phenyl ring of the PUSC scaffold and created metal coordination with Cu400 and salt bridge formation with Cu401 using the 4-hydroxyl of its β-phenyl ring. Interestingly, His263 participated in π–π stacking with kojic acid and analog **2**. Docking results of analog **3** showed two interactions, that is, a π–π stacking interaction with His259 (not His263) and hydrogen bond formation with Glu256 using the 2-hydroxyl group of the β-phenyl ring. Interactions of these ligands with the active site of tyrosinase provided the following binding energies: kojic acid = −4.2 kcal/mol, analog **2** = −5.2 kcal/mol, and analog **3** = −3.1 kcal/mol. If analogs **2** and **3** are competitive inhibitors that bind only at the active site of tyrosinase, then the higher binding affinity of **2** by docking simulation should result in more potent tyrosinase inhibition than analog **3**. However, analog **3** more potently inhibited tyrosinase than analog **2**, which suggested that **2** and/or **3** are not competitive inhibitors. Therefore, we investigated the modes of action of both using a kinetic approach (See Section 3.4). In addition, we used AutoDock Vina 1.2.0 (The Scripps Research Institute, La Jolla, CA, USA) to investigate the allosteric site of mushroom tyrosinase that binds analog **3**. Docking simulation results showed analog **3** (magenta) bound to the allosteric site with a high binding affinity (−8.1 kcal/mol) (Figure 3A), and that it bound to the same site as cinnamic acid (cyan) [54]. Thus, it seems the binding affinity of analog **3** for the active site of tyrosinase was low but has a high binding affinity for an allosteric site. These docking simulation results suggest that analog **3** is a non-competitive tyrosinase inhibitor. According to pharmacophore analysis using LigandScout, analog **3** interacts with tyrosinase at an allosteric site, as depicted in Figure 3B. More specifically, the thioxothiazolidinone ring of **3** interacted with Lys376 by π–π stacking, and the two hydroxyl substituents on the β-phenyl ring of the PUSC scaffold formed two hydrogen bonds, that is, the 2- and 4-hydroxyls interacted with Asp354 and Asp353, respectively. These two strong hydrogen bonds seemed to contribute to the high binding affinity of analog **3** for the allosteric site. Three-dimensional interactions between **3** and tyrosinase are shown in Figure 3C.

#### 3.3.2. Docking Simulations of Analogs **2** and **3** with Human Tyrosinase

Because the crystal structure of human tyrosinase has not been elucidated, a homology model was generated based on the TRP-1 amino acid sequence using Schrödinger Suite (2020-2) and the Swiss Model online server. To predict binding interactions between analogs **2** and **3** and human tyrosinase, in silico docking simulations were performed using the homology model. The software provided docking results for analog **3** and kojic acid, but not for analog **2**. Results of the docking simulations are shown in Figure 4.

Kojic acid interacted with human tyrosinase by two π–π stacking interactions between its 4-pyranone ring and the imidazole rings of histidine residues His363 and His367, by hydrogen bonding with Asn364 using its 2-hydroxymethyl group, and by forming two salt bridges with Zn6 and Zn7 using its 5-hydroxyl group. On the other hand, analog **3** hydrogen bonded with Glu203 using its 2-hydroxyl group on the β-phenyl ring of the PUSC scaffold and formed a π–π stacking interaction with Phe347 using its β-phenyl ring. These interactions provided kojic acid and analog **3** with binding energies of −4.65 and −3.09 kcal/mol, respectively. Docking results using the human tyrosinase homology model predicted that analog **3** inhibits human tyrosinase less effectively than kojic acid.

### 3.4. Kinetic Study of Analogs ***2*** and ***3***

To determine why analog **3** had lower binding affinity for the active site of tyrosinase than analog **2** but inhibited tyrosinase more than **2** as determined by IC values, inhibition mechanisms were investigated kinetically using mushroom tyrosinase in the presence of l-tyrosine. Lineweaver–Burk double reciprocal plots were used to analyze enzyme’s inhibitory mechanisms. Kinetic studies were conducted using four different concentrations of analogs **2** (0, 5, 10, or 20 μM) and **3** (0, 1, 2, or 4 μM) in the presence of l-tyrosine at five different concentrations (1, 2, 4, 8, or 16 mM). The results of the kinetic studies are provided in Figure 5, and Lineweaver–Burk plots for **2** and **3** are presented in Figure 5A,C, respectively. Lineweaver–Burk plots for **2** produced four linear lines with different slopes that merged at one point on the *y*-axis, indicating that V_max_ (3.43 × 10^−3^ mM/min) was independent of analog **2** concentration and that analog **2** competitively inhibits mushroom tyrosinase activity. Lineweaver–Burk plots for **3** also exhibited four linear lines with different slopes, but these met at one point on the *x*-axis, indicating that K_M_ (2.81 mM) was independent of analog **3** concentration and that V_max_ decreased on increasing analog **3** concentration. These results suggest that analog **3** is a non-competitive tyrosinase inhibitor, and that analog **2** binds strongly to the active site of mushroom tyrosinase, whereas analog **3** binds strongly to the allosteric site in tyrosinase, which supports our in silico docking simulation results (Figure 2 and Figure 3). The binding affinities of analogs **2** and **3** were also investigated using Dixon plots, viz. plots of the reciprocal of enzyme reaction velocity versus inhibitor concentration, which provide a graphical means of determining inhibition constants (K_i_) of enzyme–inhibitor complexes. Dixon plots of **2** and **3** are depicted in Figure 5B,D, respectively. Four linear lines with different slopes converged in the second quadrant of each Dixon plot. The mushroom tyrosinase K_i_ values of analogs **2** and **3** obtained from the *x*-axis values of each plot were 2.22 and 1.27 µM, respectively. As was expected, Dixon plots showed that analog **3** binds more strongly to mushroom tyrosinase than analog **2**.

To summarize, analog **3**, a non-competitive inhibitor, was found to bind to an allosteric site of mushroom tyrosinase more strongly than analog **2** bound to its active site, as determined kinetically and by docking simulation (Figure 2).

### 3.5. Cytotoxicities of Analogs ***2*** and ***3***

Since analogs **2** and **3** had potent inhibitory effects on mushroom tyrosinase, they were subjected to B16F10 murine melanoma cytotoxicity assays. Initially, the cytotoxicities of **2** and **3** were investigated at concentrations of 0, 1, 2, 5, 10, and 20 µM using the EZ-Cytox cell viability assay (EZ-3000; DoGenBio, Seoul, Korea). Cell viability was measured after treatment for 24, 48, or 72 h (Figure 6).

Analog **2** with a 3,4-dihydroxyphenyl (catechol) moiety was cytotoxic at 20 µM for 24 h and cytotoxic at all concentrations at 48 h and 72 h. On the other hand, analog **3** was not cytotoxic at any concentration or time. Therefore, we investigated the effects of analog **3** on cellular tyrosinase and melanogenesis in B16F10 cells.

### 3.6. Effects of Analog ***3*** on Melanin Production and Tyrosinase Activity in B16F10 Cells

B16F10 murine melanoma cells were used as a cellular tyrosinase source to confirm that analog **3** (with potent mushroom tyrosinase inhibitory activity) also inhibits cellular tyrosinase. Cells were exposed to analog **3** at 0, 5, 10, or 20 µM for 1 h, and then α–MSH (1 µM) plus IBMX (200 µM) were added, and cells were incubated for a further 72 h. Cellular tyrosinase activities were determined by measuring optical densities at 475 nm. The positive control used for the experiments was kojic acid (20 µM).

Tyrosinase inhibitory activity results are provided in Figure 7A. Co-treatment with α–MSH and IBMX increased cellular tyrosinase activity by 3.82-fold, and analog **3** significantly and dose-dependently reduced this co-treatment-induced increase in tyrosinase activity. Furthermore, analog **3** at 20 µM had greater inhibitory activity than kojic acid at the same concentration.

We also speculated that the inhibition of cellular tyrosinase activity by analog **3** would be matched by a reduction in melanogenesis. Thus, B16F10 cells were treated with analog **3** (0, 5, 10, or 20 µM) for 1 h and then stimulated with α–MSH (1 µM) plus IBMX (200 µM) for 72 h. Melanin contents were assessed by measuring optical densities at 405 nm. The positive control used for the experiments was kojic acid (20 µM).

Melanin content results are summarized in Figure 7B. α–MSH/IBMX treatment enhanced melanin contents by 3.5-fold versus non-treated controls (100%). Kojic acid at 20 µM decreased the melanin contents increased by stimulators to 3.0-fold. Analog **3** reduced α–MSH/IBMX-induced increases in melanin contents and at 5 µM had a greater effect than kojic acid at 20 µM. In particular, at 20 µM analog **3** reduced α–MSH/IBMX-induced melanin content increases to the non-treated control level. As was observed for cellular tyrosinase activities, analog **3** significantly and dose-dependently reduced α–MSH/IBMX-induced melanin content increases, which suggested that the anti-melanogenic effect of **3** is mainly due to cellular tyrosinase inhibition. Mushroom tyrosinase is a tetrameric enzyme located in cytosol, whereas murine tyrosinase is a glycosylated enzyme bound to melanosome membranes [55,56]. In addition, the amino acid sequences of these tyrosinases differ, and because of these differences, some inhibitors of mushroom tyrosinase do not inhibit murine tyrosinase. However, according to the above results, analog **3** effectively inhibited both tyrosinases. Furthermore, although analog **3** had cellular tyrosinase inhibitory activity similar to that of kojic acid, it inhibited melanogenesis to a greater extent than kojic acid. Therefore, we investigated whether another mechanism might contribute to the anti-melanogenic effect of analog **3**.

### 3.7. Effects of Analogs ***1**–**8*** on DPPH and ABTS Radical Scavenging Activities

There is a possibility that compounds with antioxidant properties inhibit the oxidations of tyrosinase substrates, such as l-tyrosine and l-dopa, without interacting with the tyrosinase [57,58]. Melanin production requires many oxidative processes, including the conversion of l-tyrosine to dopaquinone via l-dopa. Therefore, compounds with antioxidant activity have the potential to inhibit melanogenesis. DPPH (2,2-diphenyl-1-picrylhydrazyl) contains a free radical, and DPPH assays are commonly used to assess antioxidant activity. In this experiment, l-ascorbic acid (an antioxidant) was used as a positive control. The assay was conducted at a test sample and l-ascorbic acid concentration of 500 µM and a DPPH concentration of 0.18 mM. After standing test sample/DPPH mixtures for 30 min in the dark, optical densities were measured to evaluate radical scavenging activities. Results are summarized in Figure 8. l-Ascorbic acid potently inhibited DPPH radical scavenging activity by 96.8%. Of the eight analogs, **2**, **3**, and **4** strongly inhibited DPPH radical scavenging activities; the other five had weak inhibitory effects. Analog **2** with a catechol (3,4-dihydroxyphenyl) moiety was the most potent DPPH radical scavenger with 92% inhibition, which is in line with previous reports [7,13,59]. Analog **3** with a resorcinol (2,4-dihydroxyphenyl) moiety and analog **4** with a 4-hydroxy-3-methoxyphenyl moiety strongly inhibited DPPH radical activity by 73 and 72%, respectively.

The ABTS (2,2′-azino-bis(3-ethylbenzothiazoline-6-sulfonic acid)) cation radical scavenging assay is also commonly used to evaluate antioxidant effects. ABTS is converted to the ABTS radical cation by the transfer of one electron to potassium persulfate. Trolox was used as the positive control, and assays were performed at a test sample (analogs **1**–**8** and Trolox) concentration of 100 µM. The ABTS radical solution obtained by a reaction of ABTS with potassium persulfate was mixed with test samples and left in the dark for 2 min. ABTS radical scavenging abilities were evaluated by measuring absorbances at 734 nm. Results are summarized in Figure 9. Trolox scavenged 79.8% of the ABTS radicals generated. Analog **2** most potently inhibited ABTS radical activity (81.0%) followed by analog **3** (72.1% scavenging). Analog **4** (containing a 4-hydroxy-3-methoxyphenyl group) also potently scavenged the ABTS radical (57.7%), and analogs **5** and **6** (with a 3-hydroxy-4-methoxyphenyl and a 4-hydroxy-3,5-dimethoxyphenyl group, respectively), moderately scavenged the ABTS radical (**5**: 43.5%, and **6**: 46.9%). On the other hand, analog **1** with a 4-hydroxyphenyl group, and analogs **7** and **8** with bromo substituents on the β-phenyl ring weakly scavenged the ABTS radical (16.6–24.8% scavenging). Thus, of the eight analogs, five showed moderate to strong ABTS radical scavenging activities. Furthermore, the results for ABTS and DPPH radical scavenging activities were similar.

### 3.8. ROS Scavenging Effects of Analogs ***1**–**8***

Since it has been reported that reactive oxygen species (ROS) are concerned in the regulation of melanogenesis [60], we examined the ROS scavenging activities of analogs **1**–**8** using DCFH-DA (2′,7′-dichlorodihydrofluorescein diacetate) and esterase. Trolox and SIN–1 (3-morpholinosydnonimine) were used as the positive control and a ROS generator, respectively. Hydrolysis of DCFA-DA by esterase generates DCFH (2′,7′-dichlorodihydrofluorescein), which reacts with ROS generated by SIN–1 to produce the fluorescent compound DCF (2′,7′-dichlorofluorescein). Thus, ROS scavenging activities of analogs were assessed by measuring DCF fluorescence. Results are summarized in Figure 10. SIN–1 (10 µM) greatly increased ROS levels. Test samples including Trolox were treated at three different concentrations (5, 10, or 20 µM). Analogs **1**, **2**, and **3** and Trolox significantly and concentration-dependently scavenged ROS, and analogs **2** and **3** scavenged ROS more potently than Trolox.

### 3.9. Effects of Analogs ***1**–**8*** on Peroxynitrite Scavenging Activity

Several RNS (reactive nitrogen species), including peroxynitrite (ONOO^–^) and nitric oxide, are known to be involved in the induction of the melanogenesis [61,62]. Therefore, we investigated the effects of analogs **1**–**8** on peroxynitrite scavenging activity using DHR123 (dihydrorhodamine) as a ROS indicator and SIN–1 as an RNS generator. Peroxynitrite generated by SIN–1 oxidizes DHR123 to rhodamine 123, a fluorescent compound. Thus, the effects of analogs on peroxynitrite scavenging ability were determined by measuring rhodamine 123 fluorescence. Analog scavenging activities are summarized in Figure 11. SIN–1 greatly increased the concentration of peroxynitrite as compared to the non-treated control and treatment with penicillamine (the positive control) significantly and concentration-dependently reduced SIN-1-induced peroxynitrite levels. All eight analogs significantly and concentration-dependently scavenged peroxynitrite, and three analogs (**2**, **3**, and **6**) scavenged peroxynitrite more potently than penicillamine. In particular, analogs **2** and **3** scavenged peroxynitrite much more potently than penicillamine.

These results show analogs **2** and **3** are potent antioxidants that scavenge DPPH, ABTS, ROS, and RNS, and that their antioxidant effects contribute to their anti-melanogenic effects.

### 3.10. Effect of Analog ***3*** on Melanogenesis-Related Proteins and Genes Expression in B16F10 Melanoma Cells Stimulated with α–MSH Plus IBMX

To further investigate the anti-melanogenesis mechanism of analog **3**, we examined its effect on the expressions of melanogenesis-associated genes and levels of melanogenesis-associated proteins in B16F10 cells. According to a previous report [63], cellular tyrosinase levels positively influence melanogenesis in B16F10 melanoma cells, and thus, we investigated the effect of analog **3** on cellular tyrosinase levels using B16F10 cells stimulated with α–MSH/IBMX (α–MSH (1 µM) plus IBMX (200 µM)). Tyrosinase levels in the cytosol were determined by Western blotting, and the results are shown in Figure 12A. Analog **3** significantly and concentration-dependently reduced the α–MSH/IBMX-induced increase in tyrosinase expression. In addition, because MITF is a transcription factor of *tyrosinase*, *TRP-1*, and *TRP-2*, we examined the influence of analog **3** on MITF protein levels. Nuclear levels of MITF protein were measured by Western blotting in α–MSH/IBMX-stimulated B16F10 cells. As was observed for tyrosinase levels, analog **3** significantly and concentration-dependently suppressed α–MSH/IBMX-induced increases in MITF levels. These results suggest that analog **3** exerts its anti-melanogenesis effect by reducing MITF levels (and thus tyrosinase levels) and by direct inhibiting tyrosinase activity. Since MITF regulates the expression of genes responsible for melanogenesis, we used RT-qPCR to investigate whether analog **3** interfered with the expressions of melanogenesis-associated genes. Results are summarized in Figure 12B. Stimulation of B16F10 cells using α–MSH/IBMX enhanced the expressions of *tyrosinase* and *TRP-1* by 3.3- and 2.2-fold, respectively, and at 5 µM analog **3** reduced these increases to 2.6- and 1.3-fold, respectively. Furthermore, analog **3** significantly and concentration-dependently decreased the expressions of *tyrosinase* and *TRP-1*. These results suggest that the observed anti-melanogenic effect of analog **3** contributes to the suppressions of melanogenesis-related proteins and genes.

## 4. Conclusions

Eight (*Z*)-5-(substituted benzylidene)-3-cyclohexyl-2-thioxothiazolidin-4-one analogs were produced by inserting a dithionate moiety into cinnamamide derivatives. All contained the PUSC scaffold, which has been shown to confer tyrosinase inhibitory activity. Two analogs **2** and **3** potently inhibited mushroom tyrosinase 5- and 25-fold more than kojic acid, respectively. Lineweaver–Burk plots obtained in the presence of l-tyrosine as substrate showed that analogs **2** and **3** competitively and non-competitively, respectively, inhibited mushroom tyrosinase. Furthermore, these kinetic results were supported by docking simulation results using two software packages. Analog **3** inhibited melanin production in B16F10 cells in a manner analogous to its inhibition of cellular tyrosinase, which implied that the inhibition of melanin biosynthesis by **3** was mainly due to its cellular tyrosinase inhibitory activity. Our results indicate that the potent inhibitions of ROS and ONOO^–^ and of DPPH and ABTS radicals by analog **3** contribute to its inhibitory effect on melanin production. In addition, analog **3** suppressed the expressions of melanogenesis-related genes and proteins. The study shows that (*Z*)-5-(substituted benzylidene)-3-cyclohexyl-2-thioxothiazolidin-4-one analogs containing the PUSC scaffold include potential anti-melanogenic agents and that this is due to their direct tyrosinase inhibitory activities, potent antioxidant effects, and inhibitions of the expressions of melanogenesis-related proteins and genes.

## Data Availability

Data are contained within the article and Supplementary Material.

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
