# Peer review of "Design and Synthesis of (Z)-5-(Substituted benzylidene)-3-cyclohexyl-2-thioxothiazolidin-4-one Analogues as Anti-Tyrosinase and Antioxidant Compounds: In Vitro and In Silico Insights"

_antioxidants, 2022, doi:10.3390/antiox11101918_

Round 1

Reviewer 1 Report

This manuscript describes the development of anti-tyrosinase and antioxidant compounds based on benzylidene-cyclohexyl-thioxothiazolidin analogs. Many experiments including compound synthesis, enzyme inhibition assays and docking analysis were thoroughly done, and this manuscript is well written. The present results are considered to be a valuable finding, and can arouse readers’ interest. Therefore, the reviewer recommends acceptance of this paper as it is.

Author Response

This manuscript describes the development of anti-tyrosinase and antioxidant compounds based on benzylidene-cyclohexyl-thioxothiazolidin analogs. Many experiments including compound synthesis, enzyme inhibition assays and docking analysis were thoroughly done, and this manuscript is well written. The present results are considered to be a valuable finding, and can arouse readers’ interest. Therefore, the reviewer recommends acceptance of this paper as it is.

Thank you for your valuable evaluation.

Reviewer 2 Report

Excellent work

I would like a small paragraph relating the anti-oxidant activity and the anti-melanogenic activities (Ref 57,58)

A more accurate system woub more appropriate for the level of the article in regard of calculating logP 

It needs correction (twice the no of ref)

58. 58. Li, Q.; Mo, J.; Xiong, B.; Liao, Q.; Chen, Y.; Wang, Y.; Xing, S.; He, S.; Lyu, W.; Zhang, N.; et al. Discovery of Resorcinol-1019 Based Polycyclic Structures as Tyrosinase Inhibitors for Treatment of Parkinson’s Disease. ACS Chem. Neurosci. 2022, 13, 1020 81-96.

Author Response

Excellent work

I would like a small paragraph relating the anti-oxidant activity and the anti-melanogenic activities (Ref 57,58)

Thank you for your valuable suggestion. We have added a small paragraph relating the anti-oxidant activity and the anti-melanogenic activities, as suggested by the reviewer.

A more accurate system woub more appropriate for the level of the article in regard of calculating logP 

Thank you for your valuable suggestion. I’m sorry, but we are not able to provide more precise log P values because we don’t have other software to calculate log P. Please understand this situation.

It needs correction (twice the no of ref)

  1. 58. Li, Q.; Mo, J.; Xiong, B.; Liao, Q.; Chen, Y.; Wang, Y.; Xing, S.; He, S.; Lyu, W.; Zhang, N.; et al. Discovery of Resorcinol-1019 Based Polycyclic Structures as Tyrosinase Inhibitors for Treatment of Parkinson’s Disease. ACS Chem. Neurosci. 2022, 13, 1020 81-96.

Thank you for pointing out the errors. We have revised duplicate reference numbers.

Reviewer 3 Report

The manuscript by Ko et al. describes the synthesis, in silico evaluation, and experimental validation of a series of inhibitors of anti-tyrosinase compounds with use as antioxidants. The work is well conducted and results are certainly interesting. With a few minor issues solved, I would recommend publication.

My first and main concern is related to the quality and informative level of Figures. In figure 1, there is a long and poorly informative description of questions which should be reported in the main text.

Scheme 1 is confusing. A general reaction scheme should be provided, and the part "used benzaldehydes" should be removed. R groups could be listed in the reaction scheme without causing the reader to refer to precursor (that are actually numbered by letters and not with numbers).

Figure 2 is unnecessary. This kindo of J are quite standard and there is no need to evidence them. The same stands for figure 3 which should be moved in supplementary material.

Figure 4 is redundant compared with what reported in the text. It should be removed.

Additionally, the evaluation of logP with the use of a software (chemdraw) without any experimental determination is not convincing, and the following statement regarding potentially improved skin adsorption is speculative. The authors should remove this sentence, or otherwise evaluate experimentally (there are several approaches in the literature) logP of their products.

The in silico part is convincing, while concerning antioxidant tests, I wonder why the authors choose to present their data with fluroescence units (RFU) rather than  normalizing each set of data to the control value. This would provide graphs much more pleasant and easy to read.

Author Response

The manuscript by Ko et al. describes the synthesis, in silico evaluation, and experimental validation of a series of inhibitors of anti-tyrosinase compounds with use as antioxidants. The work is well conducted and results are certainly interesting. With a few minor issues solved, I would recommend publication.

My first and main concern is related to the quality and informative level of Figures. In figure 1, there is a long and poorly informative description of questions which should be reported in the main text.

Thank you for your valuable comments. As the reviewer indicated, the long informative questions in Figure 1 have been removed.

Scheme 1 is confusing. A general reaction scheme should be provided, and the part "used benzaldehydes" should be removed. R groups could be listed in the reaction scheme without causing the reader to refer to precursor (that are actually numbered by letters and not with numbers).

Thank you for your suggestion. As the reviewer suggested, the part “used benzaldehydes” was deleted.

Figure 2 is unnecessary. This kindo of J are quite standard and there is no need to evidence them. The same stands for figure 3 which should be moved in supplementary material.

Thank you for your valuable suggestion. Figure 2 has been deleted and Figure 3 has been transferred to supplementary material as suggested by the reviewer.

Figure 4 is redundant compared with what reported in the text. It should be removed.

Thank you for your valuable suggestion. Figure 4 was also deleted as suggested by the reviewer.

Additionally, the evaluation of logP with the use of a software (chemdraw) without any experimental determination is not convincing, and the following statement regarding potentially improved skin adsorption is speculative. The authors should remove this sentence, or otherwise evaluate experimentally (there are several approaches in the literature) logP of their products.

Thank you for your valuable suggestion. We deleted the sentence “These results suggest that all synthesized analogues 18 are better absorbed by skin than kojic acid.”, as suggested by the reviewer.

The in silico part is convincing, while concerning antioxidant tests, I wonder why the authors choose to present their data with fluroescence units (RFU) rather than  normalizing each set of data to the control value. This would provide graphs much more pleasant and easy to read.

Thank you for your valuable comment. The principle of the ROS scavenging activity assay is based on the reaction of H2DCF with ROS to form a fluorescent DCF, oxidized DCF fluorescence. The fluorescence intensity of DCF rapidly decreased within 30 min. Therefore, we measured the fluorescence every 5 min for 30 min. To improve the accuracy of experimental results, we used the fluorescence changes of time interval to show the moderate to high fluorescence changes per minute. In other words, the fluorescence changes of time interval to show no or low fluorescence changes per minute were not adopted for ROS scavenging activity results to improve the accuracy of experimental results.